# Active School Commuting in School Children: A Narrative Review of Current Evidence and Future Research Implications

**DOI:** 10.3390/ijerph20206929

**Published:** 2023-10-16

**Authors:** Ho Yeung Lam, Sisitha Jayasinghe, Kiran D. K. Ahuja, Andrew P. Hills

**Affiliations:** School of Health Sciences, College of Health and Medicine, University of Tasmania, Launceston, TAS 7250, Australia; sisitha.jayasinghe@utas.edu.au (S.J.); kiran.ahuja@utas.edu.au (K.D.K.A.); andrew.hills@utas.edu.au (A.P.H.)

**Keywords:** active school commuting, physical activity, active transport, active travel, children, school, narrative review

## Abstract

Active school commuting (ASC) has been proposed as a practical way to inculcate positive physical activity habits in children. This paper reviews the current evidence regarding ASC among children, highlights advances in research techniques and existing limitations in the field, and outlines future implications for research and promotion. A comprehensive literature search was conducted to identify English language studies on ASC among children aged 6–12 years, followed by a narrative review. ASC has witnessed a global decline, despite evidence of its contribution to physical activity levels. Context-dependent factors such as commuting distance and parental safety concerns are consistently identified as key determinants of ASC. Several promising interventions have been identified. Despite the limitations in intervention scope and quality, notable advancements in research techniques, such as multilevel regression and agent-based modelling, have been identified. Effective promotion of ASC to tackle childhood physical inactivity requires collaborative efforts among schools, parents, and the government, and should be tailored to address multilevel determinants within the local context. Future research should leverage recent advancements in research techniques to develop effective promotion strategies, while considering the context-dependent nature of ASC behaviours and addressing existing limitations, including the lack of standardised definitions and limited geographical and age coverage.

## 1. Introduction

Childhood physical inactivity continues to pose a significant public health challenge, with a majority of children and adolescents failing to meet the World Health Organization’s physical activity recommendations [1]. Unfortunately, the COVID-19 pandemic has likely exacerbated this issue due to social distancing restrictions and reduced access to physical activity facilities such as school, sport clubs, and swimming pools [2]. Given the propensity for habits formed in early life to transition into adulthood [3,4,5], early and middle childhood provides an opportune window to inculcate positive physical activity habits that can potentially manifest into life-long health benefits [6].

Active school commuting (ASC) was first proposed in the early 2000s as a potential source of physical activity for school children beyond the school boundaries [7,8]. With each school child typically making 360–400 trips to and from school each year [9], these journeys present an excellent opportunity to integrate physical activity into their daily routines, especially through active modes of transportation such as walking, cycling, or using non-motorised vehicles [10]. The past two decades have witnessed ASC emerging as a key area of childhood physical activity research. Nevertheless, the fact that childhood physical inactivity remains entrenched [11] makes it imperative at this juncture to examine the roles of ASC in promoting physical activity and to identify strategies to maximise its contribution in fostering a healthier and more active childhood.

This review aims to provide an overview of the current evidence in ASC. It also highlights advances in research techniques and existing limitations in the field, followed by a discussion of implications for future research and promotion.

## 2. Materials and Methods

This narrative review focuses on school children aged between 6 and 12 years, the typical age range of primary or elementary school students globally [12]. It seeks to provide a brief overview of the following key issues pertaining to ASC:What are the trends of the global prevalence of ASC?How does ASC contribute to the physical activity levels of school children?What are the health benefits of ASC?What are the determinants of ASC behaviours?What interventions are effective in increasing ASC?What are the advancements in research techniques related to ASC?What are the current limitations in ASC research?

A literature search was conducted in March 2023 across five electronic databases (PubMed, Scopus, ScienceDirect, Transport Research International Documentation (TRID), and Google Scholar) using a combination of search terms (Table 1). Additional literature was identified by reviewing the reference lists of identified articles. Furthermore, searches for ongoing studies and study protocols were conducted at trial registries, including the International Clinical Trials Registry Platform (ICTRP) and ClinicalTrials.gov. After removal of duplicates, articles were screened using the inclusion and exclusion criteria (Table 2). The full texts of the remaining papers were then reviewed and assigned under the relevant key issues. The search process is illustrated in Figure 1.

During the review, emphasis was placed on findings from systematic reviews or synthesis, supplemented by individual studies subsequently published. If a systematic review addressing a specific issue was not available, or if the existing systematic reviews did not provide sufficient information, findings from individual studies were reviewed and discussed. A total of 158 articles were included in the final review (Appendix A).

## 3. Results

### 3.1. What Are the Trends of the Global Prevalence of ASC?

Before the turn of the millennium, national longitudinal data on how school children commute to school was only available from travel behaviour surveys in a few countries, including New Zealand [13], Switzerland [14], the United Kingdom [15], and the United States [16]. However, with the growing recognition of the importance of ASC, countries like Australia [17], the Czech Republic [18], Finland [19,20,21], and Spain [22] started collecting such data as part of their national health surveys or census since the mid-2000s. Often these datasets are also supplemented by state- or city-based surveys conducted [23,24,25,26].

Taken together, the available data reveal a significant and consistent decline in the prevalence of ASC from the 1970s to the early 2000s, with only a handful of countries showing signs of plateauing reductions or improvement (Figure 2). As the decline in ASC has been observed across developed and developing countries, some researchers have considered it a global phenomenon [27].

### 3.2. How Does ASC Contribute to the Physical Activity Levels of School Children?

Numerous studies have explored the impact of ASC on physical activity levels in school children with various methods including accelerometers, pedometers, physical activity diaries, or self-reported questionnaires. Systematic reviews of these studies concluded that ASC is associated with increased total daily physical activity levels [30,31,32,33]. The majority of evidence comes from cross-sectional studies which gathered the information at one point in time. Accelerometry-based studies reported that actively commuting children accumulate an additional 3.3 to 40 min of moderate-to-vigorous physical activity (MVPA) compared to their passively commuting counterparts [34,35,36,37,38,39,40,41,42,43,44,45,46]. Similarly, pedometer-based studies have revealed that walking to and from school is associated with higher daily step counts that range from 1000 to 3400 steps [47,48,49,50,51,52,53,54,55]. Furthermore, diary-based or questionnaire studies indicated that actively commuting children are more likely to meet requirements from physical activity guidelines [56,57,58,59].

On the other hand, among the few longitudinal studies which compared the physical activity level of actively and passively commuting children across different time points, the results were mixed, with a few reporting significant differences favouring ASC at either baseline or follow-up within specific sex or age groups [60,61,62,63]. Two Swedish studies reported no difference at both baseline and follow-up at 1 and 2 years [64,65]. While reasons behind these inconsistencies were not explored, one longitudinal study reported a noteworthy finding that a switch from passive to active commuting was associated with an increase in daily minutes of MVPA [66]. This was further supported by several interventional studies on walking school bus programmes, most of which have reported increases in physical activity level for school children assigned to walk to or from their schools [67,68,69].

Most of the above studies combined walking and cycling as one single “active commuting” category in their data collection and analysis. Only a small number of cross-sectional studies compared the level of physical activity between children who walked and cycled and reported inconsistent findings. One accelerometer-based study in Denmark found that children who walked to school recorded higher values for MVPA than those who cycled, but the differences were not statistically significant [70]. In contrast, one questionnaire-based study in the United Kingdom found that cycling is associated with higher physical activity scores than walking [71].

It has been proposed that the contribution of ASC to the daily total physical activity level can operate through two mechanisms: direct and indirect [31]. The direct mechanism involves children gaining physical activity directly through the active commuting itself, while the indirect mechanism encompasses a spill-over effect, where ASC fosters a positive attitude toward physical activity, leading children to seek out other opportunities for active play and exercise outside the commuting journeys. The presence of the direct mechanism is supported by several observations. Firstly, the differences in physical activity levels between actively and passively commuting children were found to be more pronounced on weekdays and less noticeable on weekends, suggesting that ASC contributes significantly to daily physical activity on school days [35,72]. Secondly, most studies comparing physical activity levels during the school commuting journeys consistently found differences between actively and passively commuting children. In contrast, the evidence pertaining to the indirect mechanism is less conclusive, with most studies being unable to identify significant differences in physical activity levels during school hours or free time between children who actively commute and those who do not [35,38,57,73,74,75,76,77]. These findings suggest that while ASC contributes positively to physical activity level, this impact is confined only to the school commuting journeys and may not translate to contexts outside these time periods.

### 3.3. What Are the Health Benefits of ASC?

Numerous studies have examined the health benefits of ASC in children, with mixed findings across various health indicators. Most were of cross-sectional design which measured children at a single point in time. Regarding adiposity measures. while some cross-sectional studies demonstrated associations with lower body mass index (BMI) and skin fold thickness in actively commuting children, most longitudinal studies, with follow-up periods ranging from 10 weeks to 6 years, did not observe the same beneficial effect [30,31,32,33,37,60,63,78,79,80,81,82,83,84]. Cross-sectional studies which directly compared the BMI between children who walk and cycle to school were also unable to detect any significant between-group difference [46,71,85,86,87]. Similarly, studies measuring waist circumference [37,52,62,63,81,88,89,90,91] and those that utilised more reliable assessments of body composition, such as dual-energy X-ray absorptiometry [64,65], bioimpedance analysis [81,92], and air displacement plethysmography [75], did not find any discernible difference between active and passive school commuters.

Meanwhile, studies have consistently shown that children who cycle to school have a higher cardiovascular fitness level when compared with non-cyclists, including those who commute by walking or by car [70,87,93,94,95,96]. However, this relationship is not as evident between active and passive school commuters [63,71,81,88,89,97,98,99,100,101]. Findings for other measures of muscular fitness, such as hand-grip strength and standing jump, have also been inconsistent [63,87,92,101,102,103,104].

Most studies on cognitive skills reported no significant differences between active and passive school commuters [81,105,106], except for better spatial cognition in children who walked to school [107,108,109]. Similarly, studies on academic achievement are limited and have yielded mixed results [110,111,112,113]. On the other hand, ASC has been associated with fewer depressive symptoms [90] and better psychological well-being [114]. Studies on other health benefits, such as bone health and risk indicators for metabolic syndrome and cardiovascular diseases, have not yielded conclusive results [62,64,65,115,116,117].

### 3.4. What Are the Determinants of ASC Behaviours?

Like many other health behaviours, ASC is influenced by a variety of factors. While the objective built environment was traditionally considered the primary determinant [7], research adopting behavioural approaches, such as the social ecological model, has identified additional critical determinants from individual, social, and policy levels [118,119]. McMillan’s framework of the Elementary-Aged Child’s Travel Behaviour was the first to address active school commuting in school children. While acknowledging the link between objective built environment and school commuting, this framework emphasised the role of parents as the principal decisionmakers regarding the active transport behaviour of children. As such, parents’ perceptions of the commuting environment and household transportation options during the decision-making process cannot be underestimated [120] (Figure 3).

The impact of these determinants varies significantly across different study locations. Nevertheless, some consistent findings have emerged from the literature [119,121,122,123,124,125,126,127] as outlined in Table 3. For instance, positive associations were found between active commuting and environmental features like walkability and presence of physical activity facilities, while longer school journey distance and higher household car ownership had negative effects. Parental concerns regarding crime and traffic safety also play a critical role in shaping commuting behaviour. Several sociodemographic factors, including ethnicity, parental education, income levels, and neighbourhood socioeconomic status, were also associated with ASC behaviour.

Two systematic reviews on parental barriers of ASC have identified several factors that parents consider during the decision-making process, such as the lack of time to accompany the children to walk or cycle to school, ease of dropping off children on the way to work by car, and heaviness of their children’s backpacks [119,124]. Nevertheless, most of these findings were from the United States, thus limiting their generalisation. Qualitative studies have provided further insights into the parental decision-making process. Parents weighed the benefits of active commuting against safety concerns, distance, and practicality [128,129,130,131,132], while children’s perspectives highlighted the negotiation of risks, enjoyment, social interactions, and exploration during the journey [133,134].

On the other hand, the role of school in facilitating ASC appears to be under-explored in the existing literature. A systematic review of parental perceived barriers has identified four studies conducted in the United States which showed that parental perception of encouragement of ASC by school and availability of a school bus service were positively and negatively associated with ASC, respectively [124,135,136,137,138].

**Table 3 ijerph-20-06929-t003:** Key determinants of ASC behaviours.

Type of Evidence	Categories	Factors Identified	Key Findings	References
Quantitative	Objective built environment	Walkability	Higher neighbourhood walkability is positively associated with ASC	[121,122,126]
		Residential density	Higher residential density is positively associated with walking to school	[121,122]
		Land use mix	Increased land use mix diversity is positively associated with walking to school	[121,122]
		Density of physical activity facilities in the neighbourhood	Associated with a higher rate of ASC among school children	[121,122]
		Locality	Neighbourhoods located in cities or urban regions have a higher rate of ASC compared with their rural counterparts	[121]
	Household transportation options	Distance of school journey	Increased distance negatively correlated with ASC; the distance threshold varies depending on population and country	[123,125,126]
		Car ownership within the household	Associated with a decreased likelihood of children engaging in ASC	[121]
	Perceived commuting environment by parents	Perceived distance of the school journey	Identified as a major obstacle to active commuting	[119,123,124]
		Traffic safety concerns (e.g., heavy traffic, fast-moving vehicles, unsafe driving behaviours)	Associated with lower odds of active commuting to school	[119,123,124]
		Crime-related safety concerns	Associated with lower odds of active commuting to school	[119,123,124]
		Social support (presence of other children and adults) along the commute route	Associated with higher odds of active commuting to school	[124]
		Perceived neighbourhood walkability	Associated with higher odds of active commuting to school	[124]
		Discrepancy between subjective perception and objective reality	Lack of correlation between parents’ perception of neighbourhood safety and actual crime and traffic crash rate	[136,139]
	Sociodemographic factors	Sex	ASC is less prevalent in girls than in boys, with boys more likely to cycle to school while girls prefer walking	[140]
		Age	Conflicting results on the association between ASC and the age of the child, with most studies demonstrating null effect	[141]
		Household income	Children from families with higher household income are less likely to engage in active commuting to school	[125]
		Ethnicity	Children from non-white ethnic groups are less likely to engage in active commuting to school	[125]
		Parental education	Children of parents with higher levels of education are less likely to actively commute to school	[125]
		Family structure	Children with a single parent are less likely to commute actively to school, while those who are only children are more likely to do so	[121]
		Neighbourhood socioeconomic status	Neighbourhood with a higher socioeconomic status is weakly positively associated with a higher rate of ASC	[127]
Qualitative		Parents’ decision making on mode of school commute	Parents’ attitudes towards ASC are shaped by a negotiation process. They balance the benefits of active commuting with concerns about traffic and their children’s safety, along with other factors like distance to school, child’s maturity, and time constraints Regular ASC by children reduces parents’ perceived barriers	[128,129,130,131,142]
		Children’s perspective	Children may feel vulnerable during the ASC journey but are able to manage the risks. They find the commute journey enjoyable as it allows for social interactions and exploration of the environment	[134]

### 3.5. What Interventions Are Effective in Increasing ASC?

A growing body of studies have investigated the effectiveness of various types of intervention, including safe routes to school (SRTS) [7,143], walking school bus (WSB) [144], bicycle trains, and school travel plans [145], on promoting ASC among school children. As assessed by several systematic reviews, most of these interventions can result in increased ASC and physical activity among children, albeit with small effect sizes [144,146,147,148,149,150,151]. Nevertheless, their benefits can be significant at the population level [148], and economic analyses have further supported their cost-effectiveness [152].

Although the evidence is limited, several key factors may have contributed to the success of these interventions. Interventions which incorporate both educational activities and infrastructure modifications [148], and those with a stronger involvement from schools and parents, have shown greater effectiveness [146]. Interventions which targeted ASC instead of physical activity in general are also more effective [146]. These findings are consistent with the principles of the social ecological model, which emphasises the importance of addressing key factors from multiple levels to create an enabling environment.

Several barriers that may threaten their effectiveness have also been identified. Many studies reported substantial variation of implementation across schools due to the varied degrees of motivation among schools [148]. The limited availability of funding and resources also affects the long-term sustainability of the interventions [148]. Furthermore, some initiatives which depend on volunteer participation, such as WSB programmes, face on-going difficulties in recruiting sufficient volunteers [144].

### 3.6. What Are the Advancements in Research Techniques Related to ASC?

**Increasing utilisation of geospatial data:** One notable development is the increased utilisation of geographic information systems (GISs) in measuring the objective built environment. Prior to the advent of GISs, time- and resource-intensive on-site auditing was the primary method for obtaining such information [126]. In recent years, spatial datasets of street networks, land use information, and locations of schools and physical activity facilities have become more accessible from government and non-profit organisations alongside the rapid advancement of a variety of GIS software which has increased the processing and analysis of these data drastically [153]. Furthermore, these spatial datasets can be combined with area- or neighbourhood-level data from other sources, such as census data, to generate secondary spatial datasets that capture crucial factors found to be positively associated with ASC [126]. One notable example concerns the measurement of school commute distance. While early studies relied on parental reports that are prone to inaccuracies [154], the availability of GIS enables a more precise estimation by calculating straight-line distances between home and school using spatial data [155]. However, this approach does not account for the actual commuting route taken. To address this limitation, subsequent studies have incorporated routing algorithms, such as those from ArcGIS and Google Maps, to estimate the shortest possible routes [85,156]. More recently, more accurate methods have been adopted, which include having school children draw their commuting routes on an electronic map [157] or wear GPS monitors to capture their actual route [158,159].

**Increasing utilisation of standardised instruments:** Another advancement involves the increasing use of standardised instruments which has enabled direct comparisons across studies. One notable example is the Neighbourhood Environment Walkability Scale (NEWS), a questionnaire designed to measure residents’ perceptions of the environmental attributes of their local area in relation to physical activity [160]. NEWS and its derivatives [161,162] have been employed in studies on ASC to gauge parents’ and children’s perception on their neighbourhood’s walkability [85,163,164,165,166]. Recently, there have also been initiatives to develop and validate standardised instruments to assess and measure school children’s ASC behaviours. Examples include the Mode and Frequency of Commuting To and From School Questionnaire [167,168] and its derivative, the Family Commuting-to-School Behaviour Questionnaire [169].

**Advancements in analytical approaches:** Early studies predominantly employed simple univariate or bivariate statistical techniques at the individual level, overlooking the multilevel nature of the interactions [119] as well as the social and spatial nesting of school children [170]. Inspired by McMillan’s framework, a number of theoretical frameworks on active school commuting, grounded in the social ecological model, have emerged in recent years [118]. These frameworks offer diverse perspectives on how parents’ decision-making processes are influenced by different factors across various levels. The availability of these different frameworks has supported the adoption of more sophisticated analytical methods, such as multilevel regression modelling, structural equation modelling, and hierarchical modelling as seen in some recent studies [157,170,171,172,173]. These approaches account for the nested and multilevel relationships between school children, parents, neighbourhoods, and schools, thus enabling more appropriate data analysis and a deeper understanding of the complex and interrelated factors that influence school commuting behaviour.

**Emerging application of modelling techniques:** The logistical complexity and resource requirements of evaluating ASC interventions under real-world settings have prompted researchers to adopt alternative modelling approaches. The logic pathway modelling to evaluate the national effects of a walking school bus intervention on body mass index and disability-adjusted life years is a good case in point [174]. Agent-based modelling (ABM) is another modelling technique gaining popularity in recent years [175]. ABM explicitly models the decision-making processes of individual parents and children regarding school commute modes and incorporates key determinants such as distance, safety concerns, and social norms. This approach allows researchers to simulate in silico different intervention scenarios and predict their impact on ASC behaviours, thus providing valuable insights for intervention design and evaluation. Several studies have utilised ABM to study ASC and demonstrated its potential for guiding future research in this field [176,177,178,179].

### 3.7. What Are the Current Limitations in ASC Research?

**Classification of ASC:** The lack of standardised classification schemes for ASC makes it difficult to compare and generalise findings across studies and localities [180]. The widely varying thresholds for classifying school children as active commuters, as well as the combination of walking and cycling as a single “active commuting” category, may have weakened the findings with determinants, physical activity levels, and health benefits and effective interventions [30]. Notably, studies that measured ASC as an ordinal or continuous variable, and those which separated walking and cycling, tend to report significant findings [36,47,59].

**Limited age and geographical coverage:** Most ASC studies focus on older children aged nine years or above, with limited representation of younger age groups. Extrapolation of findings from older school children might not be appropriate, as younger children may have different commuting patterns and constraints. For instance, they have lower level of independent mobility [33], may live closer to their schools, and will only walk when the school is nearby or with adult supervision [32]. Moreover, research on ASC is predominantly derived from metropolitan or urban areas in developed nations, with limited data from rural, regional, and developing countries [78,147].

**Heterogeneity, mediators, and confounders in study methodology:** The substantial heterogeneity among studies in terms of design (e.g., cross-sectional, longitudinal, non-randomised, and randomised controlled trials), definitions, and methodology has impeded the conduct of meta-analysis. Consequently, most systematic reviews have to rely on the questionable vote-counting technique for summarising results [181]. The wide variation of study findings across different cultures, economies, and geographical locations attests to the importance of contextual factors in shaping school commuting behaviours [122]. However, the effects of various mediators and confounders, including diet, school journey distance, ethnicity, culture, and socioeconomic status are often overlooked in existing studies [148].

**Limited research in key areas:** The scarcity of studies on the benefits of ASC, such as mental health and academic performance, may contribute to the limited appeal of ASC among parents and policymakers [30,31,32,33,78,79,80,104,182,183]. Similarly, the existing body of qualitative literature on this topic is also limited [134], thus stymieing a more comprehensive understanding of the dynamics underlying ASC behaviours. Interventional studies also suffer from weak quality resulting from non-randomised design, lack of blinding and control groups, and reliance on self-reporting [147,148,149]. Moreover, there is also a lack of utilisation of behavioural frameworks in interventional studies, which has limited the understanding on the mechanism underlying behavioural changes and the generalisability of findings to other settings [146,147,148].

## 4. Discussion

### 4.1. Future Research Implications

Future research should aim to adopt a clearer and standardised definition of ASC, while also analysing different forms of active commuting separately [119] and as ordinal or continuous variables [180]. Researchers should broaden their study scope by targeting younger age groups and exploring ASC behaviours in rural, regional, and developing areas. They should go beyond cross-sectional studies by utilising longitudinal studies with extended follow-up periods to better understand the long-term effects on physical activity level and other health indicators. Moreover, like several recent studies [184,185], future studies may wish to adopt a mixed-methods approach, which combines quantitative and qualitative techniques [186] to generate more useful information regarding the local context within a single study. Given the complex and context-specific nature of ASC behaviours, the use of more sophisticated analytical methods, such as multilevel regression modelling and structural equation modelling, can help elucidate the effects of various mediators and confounders. This in turn can inform the development of targeted interventions and policies that address the specific needs and challenges of the local context.

Future interventional studies should focus on optimisation of study quality to generate robust evidence. This can be achieved through the utilisation of representative samples, larger sample sizes, randomised controlled trial designs, and the use of valid and reliable objective measures [148,150,151]. In addition, interventional studies should be grounded in a theoretical framework and incorporate mediators identified from behavioural studies. Furthermore, longer follow-up periods are necessary to evaluate the long-term efficacy of interventions. To overcome resource limitations, modelling techniques, such as agent-based modelling, can be utilised to plan and identify potentially effective interventions.

### 4.2. Implications for Promotion of ASC

Promoting ASC requires a comprehensive and evidence-based approach at both the national and local levels. Despite some mixed findings, the existing evidence lends support to the positive impact of ASC on increasing physical activity levels among school children. Given the current urgency of the childhood physical inactivity and obesity crisis, ASC should be recognised as a pragmatic response and an integral part of the school health policy.

To ensure their effective implementation, it is important to address the lack of prevalence data on ASC in many regions of the world. Establishing regular surveillance of ASC prevalence through national or regional health surveys becomes imperative. Such data will not only inform stakeholders about the effectiveness of current initiatives but also shape future strategies. In the absence of such a mechanism, a baseline situational analysis of the ASC in the target region is indispensable.

Schools play a pivotal role in fostering ASC habits among students. To promote ASC effectively, a context-specific approach is required at the school level. Local partnerships between government, schools, parents, and community members allow tailoring of strategies to the specific needs and challenges of their communities. While reducing school commuting distances for existing schools may be impractical [123], a multifaceted approach that includes infrastructure enhancements, provision of safe and supportive commuting environments, and educational initiatives can positively influence other modifiable determinants, such as parental safety concerns. Recognising the role of parents as major decisionmakers, highlighting the known benefits of ASC, such as improved fitness and mental health, can help counteract some of the parental barriers. Collaborative efforts are necessary to also overcome general and locally specific challenges, including resource limitations.

Equity considerations are also important to ensure all school children have access to ASC opportunities. Given the positive correlation between neighbourhood socioeconomic status, crime safety concerns, and ASC prevalence, targeted interventions that address transportation inequities and creating safe routes for walking and cycling in disadvantaged areas are needed. Nevertheless, the inverse relationship between ASC and income levels implies the possible existence of a gradient of ASC preference even in wealthier neighbourhoods. Hence, strategies are also needed to focus on the specific factors influencing ASC in different families.

## 5. Conclusions

This review offers a comprehensive overview of the current landscape of ASC research, highlighting its potentials in addressing childhood physical inactivity (Figure 4). Current evidence supports the direct contribution of ASC to physical activity levels and several health benefits. The decline in global prevalence of ASC presents an opportunity for it to be one of the strategies to promote regular physical activity among school children. Behavioural frameworks encompassing multilevel determinants have facilitated the understanding of ASC behaviours. Several promising interventions, such as the walking school bus, have also be identified. While advances in research techniques have supported the progress in ASC research, it is essential to address the existing limitations and challenges in methodology and implementation, including the lack of standardised ASC definition and paucity of research in younger age groups and rural settings.

Although future research aimed at addressing the identified research gaps is expected to be carried out mostly in developed countries, it is crucial to acknowledge that ASC remains under-examined in many developing countries and rural areas of developed countries. As ASC behaviour is highly context specific, it is prudent for researchers to first study the local situation and comprehend the local dynamics contributing to the observed ASC behaviours, using quantitative and qualitative study methods supported by relevant behavioural frameworks. This approach can help identify critical local barriers and enablers of ASC behaviour, on which targeted interventions can be developed in partnerships involving the government, schools, parents, and local community. Modelling using ABM can be considered to select the interventions with the highest potential for effectiveness.

## Figures and Tables

**Figure 1 ijerph-20-06929-f001:**
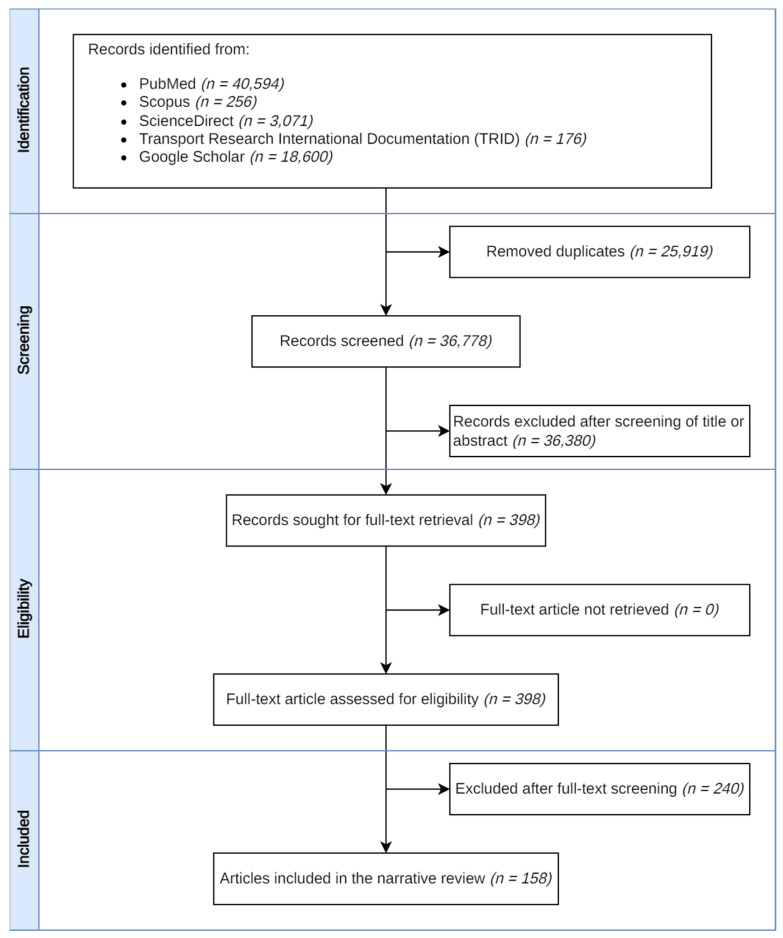
The literature search flowchart.

**Figure 2 ijerph-20-06929-f002:**
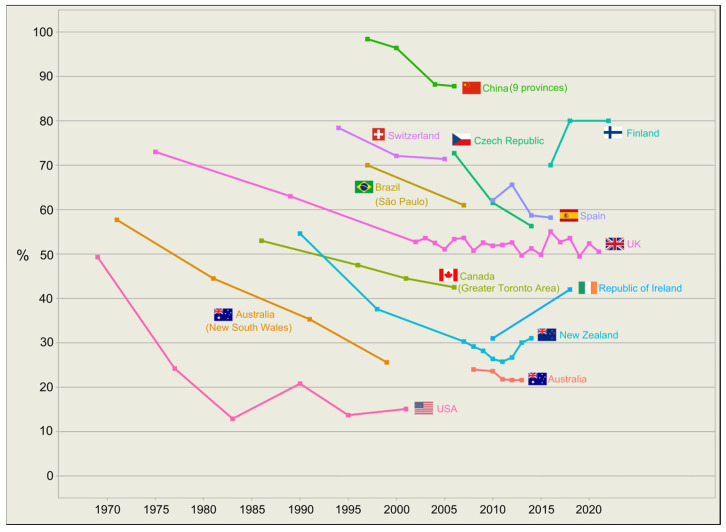
Global prevalence of ASC among primary school children. Prevalence refers to percentage of school children who commute to or from school using active modes of transportation. Furthermore, additional government data from Australia, Finland, New Zealand, the United Kingdom, and the United States have been included here as they are unavailable in the published literature. Data are included if they cover school children aged between 6 and 12. For surveys which report breakdowns of different modes of transportation, data of different “active” modes are combined to give an overall prevalence of ASC. Caution should be exercised when comparing between countries or regions, given the marked heterogeneity in how ASC is defined and measured, as well as the variations in target age groups across different surveys (further discussed in Section 3.7). Source of data (in alphabetical order): Australia—CensusAtSchool [17]; Australia (New South Wales)—Household Travel Surveys [23]; Brazil (São Paulo)—São Paulo Metropolitan Area Household Travel Survey [24]; Canada (Greater Toronto Area) [25]; China (9 provinces)—China Health and Nutrition Surveys [26]; Czech Republic—Health Behaviour in School-aged Children study [18]; Finland [19,20,21]; New Zealand—The New Zealand Household Travel Survey [13]; Republic of Ireland—The Children’s Sport Participation and Physical Activity Study [28]; Spain—PACO Study [22]; Switzerland—Swiss Microcensus on Travel Behaviour [14]; UK—UK National Travel Survey [15,29]; USA—National Household Travel Survey [16].

**Figure 3 ijerph-20-06929-f003:**
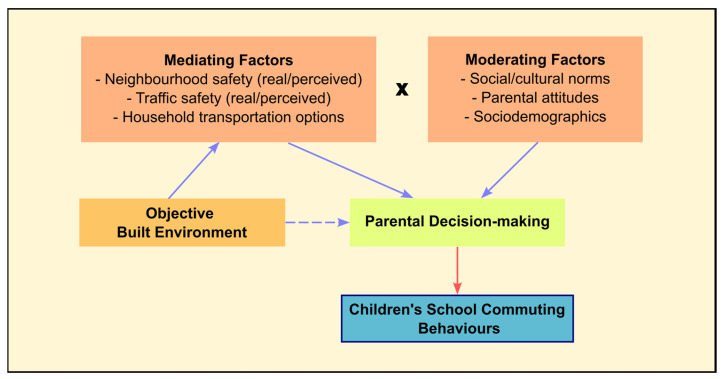
McMillan’s Conceptual Framework of an Elementary-Aged Child’s Travel Behaviour (adapted from McMillan [120]).

**Figure 4 ijerph-20-06929-f004:**
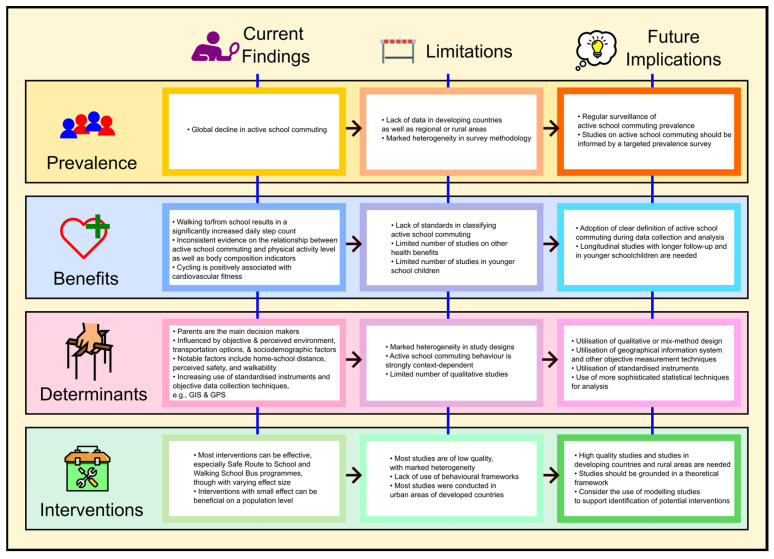
Summary of key review findings and future research implications.

**Table 1 ijerph-20-06929-t001:** Search terms/strings utilised for identifying pertinent literature.

Electronic Databases	Search Terms	Number of Articles
PubMed	#1: schoolchild OR child #2: school OR primary school OR elementary school OR junior school #3: active transport OR active commute OR active travel OR walk OR bike OR cycle OR bicycle OR skateboard OR vehicle OR motorised transport OR car OR drive #4: #1 AND #2 AND #3	40,594
Scopus	(schoolchild OR child) AND (“school” OR “primary school” OR “elementary school” OR “junior school”) AND (“active transport” OR “active commute” OR “active travel”) AND (walk OR bike OR cycle OR bicycle OR skateboard OR vehicle OR “motorised transport” OR car OR drive)	256
ScienceDirect	(schoolchild OR child) AND (school OR “primary school” OR “elementary school” OR “junior school”) AND (“active transport” OR “active commute” OR “active travel”)	3071
Transport Research International Documentation	1: schoolchild OR child 2: school OR primary school OR elementary school OR junior school 3: active transport OR active commute OR active travel OR walk OR bike OR cycle OR bicycle OR skateboard OR vehicle OR motorised transport OR car OR drive 4: 1 AND 2 AND 3	176
Google Scholar	(schoolchild OR child) AND (“school” OR “primary school” OR “elementary school” OR “junior school”) AND (“active transport” OR “active commute” OR “active travel”) AND (walk OR bike OR cycle OR bicycle OR skateboard OR vehicle OR “motorised transport” OR car OR drive)	18,600

**Table 2 ijerph-20-06929-t002:** Inclusion and exclusion criteria used for the literature review.

Inclusion Criteria	Exclusion Criteria
Publication year: Published before March 2023 Article type: Investigative studies (quantitative or qualitative), systematic reviews and synthesis, commentaries, editorials, and theses Target age group: Between 6 and 12 years Study focus: Address any of the 7 key issues pertaining to ASC below: What are the trends of the global prevalence of ASC? How does ASC contribute to the physical activity levels of school children? What are the health benefits of ASC? What are the determinants of ASC behaviours? What interventions are effective in increasing ASC? What are the advancements in research techniques related to ASC? What are the current limitations in ASC research?	Non-English literature

## Data Availability

Not applicable.

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
