# Peer review of "Active School Commuting in School Children: A Narrative Review of Current Evidence and Future Research Implications"

_ijerph, 2023, doi:10.3390/ijerph20206929_

Round 1
Reviewer 1 Report
I was pleased to read the article 'Active school commuting in school children: a narrative review of current evidence and future research implications'.
In my opinion it is very coherently and logically written.
It covers an important problem that many countries around the world are experiencing. The authors' research shows that these are mainly developed and developing countries. This can be perfectly observed in my country over a period of 50 years.
My comment would be to make the purpose of the article clearer in both the abstract and the introduction.
Reviewer 2 Report
The manuscript provides a comprehensive review of studies on active school commuting among children aged 6-12 years, offering valuable insights and outlining future research implications. Overall, I found the article to be well-structured and insightful. However, there are two minor suggestions that, if addressed, could enhance the article's visibility and clarity.
It would be beneficial to expand on the importance of the subject in the introduction section. While I understand that this is a review article, elaborating on the significance of active school commuting and why it merits attention would provide context and motivation for the reader.
Although the second section provides useful information on locating articles, it might benefit from additional explanations. For instance, the author could consider specifying their "time boundary" for the review. By indicating a particular point in time, it would clarify which articles, if any, published before that point were not included or reviewed in this study. This added detail would enhance the transparency of the review process.
Reviewer 3 Report
This is a comprehensive narrative review and well-written paper on an important topic of high relevance to the health and well-being of school children. I think a lot of important aspects have been addressed including the collection and analysis of data.
I have no major issues with the paper overall, but I have noticed a few omissions and lack of detail that I believe will need to be addressed before the paper is ready for publication.
These are outlined in the attached file.
